# Immunotherapy and Gene Therapy for Oncoviruses Infections: A Review

**DOI:** 10.3390/v13050822

**Published:** 2021-05-02

**Authors:** Nathália Alves Araújo de Almeida, Camilla Rodrigues de Almeida Ribeiro, Jéssica Vasques Raposo, Vanessa Salete de Paula

**Affiliations:** Laboratory of Molecular Virology, Oswaldo Cruz Institute, Oswaldo Cruz Foundation, 21040-360 Rio de Janeiro, Brazil; nathalia.almeida@ioc.fiocruz.br (N.A.A.d.A.); camilla_almeida@hotmail.com (C.R.d.A.R.); Jessicavasquesr@gmail.com (J.V.R.)

**Keywords:** immunotherapy, gene therapy and oncovirus

## Abstract

Immunotherapy has been shown to be highly effective in some types of cancer caused by viruses. Gene therapy involves insertion or modification of a therapeutic gene, to correct for inappropriate gene products that cause/may cause diseases. Both these types of therapy have been used as alternative ways to avoid cancers caused by oncoviruses. In this review, we summarize recent studies on immunotherapy and gene therapy including the topics of oncolytic immunotherapy, immune checkpoint inhibitors, gene replacement, antisense oligonucleotides, RNA interference, clustered regularly interspaced short palindromic repeats Clustered Regularly Interspaced Short Palindromic Repeats (CRISPR)-based gene editing, transcription activator-like effector nucleases (TALENs) and custom treatment for Epstein–Barr virus, human T-lymphotropic virus 1, hepatitis B virus, human papillomavirus, hepatitis C virus, herpesvirus associated with Kaposi’s sarcoma, Merkel cell polyomavirus, and cytomegalovirus.

## 1. Introduction

Oncoviruses contribute significantly to cancer cases worldwide. A century of studies on the presence of viruses in tumors has led to the discovery of viruses that have the ability to alter the cell cycle of infected cells, inducing the development of tumors. There are currently eight well-accepted human oncoviruses, including Epstein–Barr virus (EBV), human T-lymphotropic virus 1 (HTLV-1), hepatitis B virus (HBV), human papillomavirus (HPV), hepatitis C virus (HCV), herpesvirus associated with Kaposi’s sarcoma (KSHV) or human herpes virus-8 (HHV-8), Merkel cell polyomavirus (MCPyV), and human cytomegalovirus (HCMV) [1,2]. Most of the viruses are DNA viruses, except for HCV, which has a sense single-stranded RNA virus (ssRNA genome) and HTLV-1, which is a retrovirus. Oncoviruses are collectively responsible for 12–15% of human cancers worldwide. The discovery of oncoviruses has led to the identification of preventive and therapeutic approaches that significantly change the epidemiology of viruses associated with cancers [3].

Immunotherapy is a type of biological treatment that aims to enhance the immune system so that it can fight infections and other diseases such as cancer. In the past few decades, immunotherapy has become an important part of treating some types of cancer. New types of immunotherapeutic treatments are being developed, which will have a major impact on cancer treatment. Immunotherapeutic treatments act in different ways: some stimulate the body’s immune system in a general way, while others help the immune system specifically target cancer cells. Immunotherapy has been shown to be more effective in some types of cancer caused by viruses.

Gene therapy, also known as gene editing, is an innovative treatment that consists of a set of techniques that can serve to be useful in the treatment and prevention of complex diseases (such as genetic diseases and cancer) by modifying specific genes. This type of treatment consists of causing changes in the DNA of the cells affected by the disease and activating the body’s defenses, in order to recognize the damaged tissue and promote its elimination. Thus, the diseases that can be treated in this way are those that involve alterations in the DNA. The idea of using recombinant DNA techniques to correct the genome was initially inspired by diseases caused by mutations in a single gene (likewise known as monogenic diseases). In this case, the idea is to replace or supplement the expression of the dysfunctional gene by inserting one or more copies of the therapeutic gene [4,5,6]. Currently, gene therapy is used for oncoviruses. In certain cases, it is possible to plan an intervention through gene therapy, aiming to reduce or prevent disease progression. Interventions can be based on the knowledge of genetic determinants of susceptibility or severity, or on the opportunity to alter fundamental mechanisms of the viruses.

In this review, we will cover the main methods used in gene therapy and immunotherapy, including oncolytic immunotherapy, immune checkpoint inhibitors (ICIs), gene replacement, antisense oligonucleotides, RNA interference (RNAi), clustered regularly interspaced short palindromic repeats (CRISPR)-based gene editing, transcription activator-like effector nucleases (TALENs), and customized treatment for oncoviruses.

## 2. Immunotherapy

### 2.1. Oncolytic Immunotherapy

Oncolytic viruses (OVs) are native or recombinant viruses that kill cancer cells. These viruses cause cancer cells to die at the end of their replication cycles, either by lysis or by activation of antitumor immune response, thereby minimizing damage to the normal tissues [7,8].

Some viruses widely used as OVs include parvoviruses, Myxoma virus, Reovirus (Reo), Newcastle disease virus (NDV), Seneca valley virus, poliovirus, measles virus, virus vaccinia, adenovirus, herpes simplex virus (HSV), and vesicular stomatitis virus (VSV) [7,9,10,11,12,13,14].

Currently, there are only a few studies that have used OVs in cancers associated with oncogenic viruses, and these studies have mostly been focused on cancers associated with HBV, HCV, or HPV. Abdullahi et al. (2018) found that systemic administration of a new oncolytic hybrid vector comprising VSV and NDV, called recombinant VSV-NDV, into immunocompetent mice with orthotopic hepatocellular carcinoma (HCC) resulted in significant prolongation of survival [15].

Studies have already demonstrated that oncolytic Reo triggers innate immune activation in primary human liver tissue, in the absence of cytotoxicity and independent replication of the viral genome in HCV and HBV-associated HCC models, as well as an alternative endogenous lymphoma model associated with EBV [16,17,18,19,20,21]. In HCC, some OVs were tested in phase 1 and 2 trials [22,23,24].

For cancers associated with HPV, a study by Atherton et al. that evaluated virotherapy with Maraba virus (MG1-Maraba) MG1-E6E7 demonstrated that MG1-Maraba can acutely alter the tumor microenvironment in vivo, and therefore may be suitable as an oncolytic treatment against clinical cancer related to HPV [25,26]. A study by Keshavarz et al. (2020), in which NDV was delivered using a mesenchymal stem cell engineering system, suggested it as a potentially effective strategy for cancer immunotherapy associated with HPV, as it induced splenic Th1 immune responses and apoptosis in the tumor microenvironment. Another study by Suksanpaisan et al. (2018) demonstrated NDV an excellent platform for oncolytic immunotherapy of tumors that express HPV target antigens, as it offered an appropriate balance of efficacy and safety in first-phase clinical trials in humans [27,28].

### 2.2. Immune Checkpoint Inhibitors

During the process of infection of the host cell and establishment of persistent infection, carcinogenic viruses develop immune escape mechanisms such as production of anti-inflammatory cytokines, induction of regulatory T cells, and an increase in the expression of immunological checkpoints [29].

The immunological checkpoints most studied and implemented in practice are protein 4 associated with cytotoxic T lymphocytes (CTLA-4), programmed cell death protein-1 (PD-1), and programmed cell death protein ligand-1 (PD-L1) [30]. Several other surface receptors are also being explored as therapeutic targets, including inhibitory receptor lymphocyte activation gene 3 (LAG-3), killer cell immunoglobulin-like receptor (KIR), T cell immunoreceptor with immunoglobulin (Ig) domains and T cell immunoreceptor with Ig and ITIM domains ITIM (TIGIT), and T cell immunoglobulin and mucin domain 3 (TIM-3, also known as HAVCR2) [31,32,33,34].

Viral infections are estimated to contribute 15–20% of all human cancers worldwide and represent a special entity and a subgroup that has not been fully studied [35]. These virus-induced cancers are likely to have a specific immune profile; consequently, their response is expected to be different for immune checkpoint inhibitors than for other types of cancer. Many hypotheses can explain this possible differential tumor response; higher mutational load and higher expression of PD-L1 in virus-induced cancers are the most reported [36,37,38,39,40,41]. In an article published by Schumacher and Schreiber evaluating the mutational burden in different cancers, six of the first 14 cancers with the highest mutational burden were virus-induced cancers [40].

The immune checkpoint inhibitors (ICIs) are predominantly monoclonal antibodies that have been shown to be effective against a variety of cancers [42,43]. These work by blocking the binding of molecules from the immune control point to their ligands, reversing the inactivation of T cells, increasing the immune response of T cells, and resisting external aggression, such as viral infections. Via such effects, ICIs assist in the elimination of the virus in infected patients, and thus, have a greater effect on cancers associated with viruses [29,30,31,32,33,34,35,36,37,38,39,40,41,42,43,44].

Several clinical trials have assessed the safety and efficacy of ICIs in patients infected with human immunodeficiency virus (HIV). A placebo-controlled dose-escalation study (NCT02028403) of BMS-936559 (anti-PD-L1 antibody) was conducted on six adult men infected with HIV-1, whose plasma HIV-1 RNA levels were detected as a single copy. Of the six participants who received BMS-936559, the average percentage of HIV-1 Gag-specific CD8+ T cells increased in two participants, demonstrating that single infusions of BMS-936559 appeared to increase HIV-1-specific immunity in the participants [45].

A study (NCT03407105) that evaluated the safety of multiple ascending doses of ipilimumab (anti-CTLA-4 antibody) and its ability to increase the immune response to HIV-1 in HIV-1-infected participants showed that two participants (8.3%) displayed a decrease in baseline HIV-1 RNA, while 14 participants (58.3%) displayed an increase in baseline HIV-1 RNA. Ipilimumab was well tolerated and associated with variations in HIV-1 RNA. However, the mechanisms underlying the increased variation in HIV-1 RNA were not clarified in this study [46]. In addition, two case reports of HIV-infected patients treated for cancer with anti-CTLA-4 antibody [47] or anti-PD1 antibody [48] have recently shown a good safety profile, as confirmed in two series of patients treated for melanoma (*n* = 9) [49] and lung cancer (*n* = 10) [50].

Patients with HIV infection are at increased risk for cancer, and this is the leading cause of death among non-AIDS-defining illnesses in these patients. A systematic review conducted by Cook et al. (2019) to summarize the effectiveness of ICI therapy in HIV-infected patients with cancer showed that among 34 patients with known HIV pretreatment and post-treatment burdens, HIV remained suppressed in 26 of the 28 (93%) patients with undetectable HIV burden [51]. The overall response rate (ORR) in these HIV-related patients was 30% for non-small-cell lung cancer, 27% for melanoma, and 63% for Kaposi’s sarcoma. Therefore, ICIs can be an effective treatment option in this population of HIV-1-positive patients and their associated cancers [51].

HPV is responsible for more than half of all infections attributable to cancers in women worldwide [52]. Tumors associated with HPV have been found to display strategies to escape immune attacks. Despite the widespread use of various treatment options, most HPV-related cancers are still difficult to cure worldwide. However, the emergence of ICIs in recent years has offered a new hope for the treatment of these cancers, as these display the potential to overcome immune escape in HPV-associated tumors [29,53].

The FDA approved pembrolizumab (anti-PD1 antibody) in 2018 for treatment of recurrent or metastatic cervical cancer, based on the results of the Keynote-028 trial, which reported an ORR of 17% (95% CI: 5–37%); however, the patients’ HPV infection status was not clear in this study [54]. A study (CheckMate358) [55] exploring the safety and efficacy of an anti-PD1 antibody (nivolumab) in cancers associated with viruses is currently being carried out. The study has reported that of the 24 patients treated, the ORR was 20.8% and the disease control rate (DCR) (complete response (CR) + partial response (PR) + stable disease (SD)) was 70.8%, at an average follow-up of 31 weeks. All responses were observed regardless of HPV status in patients with cervical cancer [55]. Another multicenter, single-phase, phase II clinical trial (NCI-9673/NCT02314169) also studied the therapeutic effect of nivolumab in patients with metastatic squamous cell carcinoma of the anal canal (SCCA). The response rate was 24% (95% CI = 15–33) among the 37 patients who received nivolumab. HPV was detected in all the specimens tested in this study, given its high prevalence in SCCA [56]. Therefore, nivolumab has shown an encouraging therapeutic effect in patients with HPV-associated cancer and is worthy of further evaluation in these patients [57].

Ho et al. (2018) published a case report that described a robust response to combined therapy with ICIs in HPV-related small cell cancer. A 69-year-old female patient with HPV-related head and neck (H&N) small cell cancer, an aggressive and rare variant of HPV-related H&N cancers, demonstrated a significant response to immunotherapy using a combination of ipilimumab (anti-CTLA-4 antibody) and nivolumab, after progressing with the most adopted treatment approach involving radiation and combination chemotherapy with platinum and etoposide. A robust partial response was observed within 2 months of starting immunotherapy in this case [58].

HCC is one of the most common cancers in the world and ranks third in global incidence, being a highly heterogeneous malignancy with different risk factors, including HBV and HCV. Despite recent progress in the management of HCC, the condition is mostly diagnosed in the advanced stages, when the therapeutic options are limited [59,60]. Recently, ICIs have been used to treat HCC in patients with HCV and HBV.

Studies have shown that the death of tumors by direct methods (known as ablation) can result in activation of the immune system [61,62]. ICIs can increase the efficacy of anticancer therapy by activating the immune system to recognize and eliminate residual cancerous lesions. A study (NCT01853618), which included 32 patients with HCC, aimed to demonstrate whether tremelimumab could be combined with ablation in a safe and viable way. Five of the 19 assessed patients (26.3%) achieved a confirmed partial response, while 12 of 14 patients (85.7%) with quantifiable HCV experienced a marked reduction in viral load. Therefore, tremelimumab (anti-CTLA-4 antibody), in combination with tumor ablation, may be a potential new treatment for patients with advanced HCC, leading to the accumulation of CD8^+^ T cells and a reduction in the HCV load [62].

A case report published by Fukuda et al. (2020) demonstrated a case of HCC with a decline in HCV RNA after administration of anti-PD-1 antibody (nivolumab). HCV-RNA titers decreased, without liver damage, after the initiation of treatment with nivolumab; the decline in HCV RNA titers was observed regardless of the antitumor effect [63].

In an international study (CheckMate 040/NCT01658878), multicenter, open, phase I/II trials with nivolumab in adults with advanced HCC, regardless of etiology (not amenable to curative resection or local treatment with or without previous treatment with sorafenib), a large percentage of patients were found to be infected with HBV or HCV, and nivolumab showed comparable safety and efficacy among the patients treated with sorafenib [64].

In a retrospective analysis published in 2019, of the 55 patients with HBV or HCV and advanced primary HCC who received anti-PD-1 agents (nivolumab and pembrolizumab), the median overall survival was 15 months, while the median progression-free survival (PFS) was 10 months. No patient displayed a CR, while 12 (22%) participants displayed PR, resulting in an ORR of 22%. Thirty-seven (67%) patients had SD and 6 (11%) had progressive disease (PD) in the first radiological evaluation. The DCR was 89%. The total rate of side effects was 61.8%, but most were relieved after treatment, thus demonstrating that immunotherapy directed against PD-1 is a safe and effective treatment for advanced primary HCC [65,66].

EBV is associated with several types of human neoplasms, such as malignant lymphoma, nasopharyngeal carcinoma (NPC), and gastric cancer (GC) [29]. Lymphomas and NPC associated with EBV have a virus-mediated overexpression of PD-L1, making them sensitive to PD-1 blockade as well [67,68].

According to a study reported at the 2015 European Cancer Congress, upon treatment with pembrolizumab (anti-PD-1), more than a fifth of patients with previously treated metastatic NPC showed an objective measurable response, while two-thirds of the patients in the study experienced some degree of reduction in the size of the target lesion (Hamid et al., 2013). A recent study (NCT02339558) used nivolumab (another anti-PD1 antibody) to treat nine patients with recurrent and/or metastatic NPC for more than 12 months (20%) and found the overall one-year survival rate to be 59% (95% CI = 44.3% to 78.5%) and the PFS in 1 year to be 19.3% (95% CI = 10.1% to 37.2%) [69].

Several multicenter, single-phase, phase II studies have been conducted in China for relapsed/refractory classical Hodgkin lymphoma patients associated with EBV who have failed treatment with at least two lines of chemotherapy or have had autologous stem cell transplantation. Sintilimab (anti-PD-1) was evaluated in the ORIENT-1 study with 96 patients, wherein the patients received 200 mg of synthilimumab intravenously once every 3 weeks for a maximum period of 24 months. The median duration of follow-up was 10.5 months, the ORR was 80.4% with a CR rate of 33.7%, and the PFS at 6 months was 77.6% [70]. Camrelizumab (anti-PD-1) was evaluated in the SHR-1210-II-204 study involving 75 patients who were treated with 200 mg camrelizumab intravenously every 2 weeks until disease progression or intolerable toxicity. The ORR reported by the Radiological Committee was 76% with 21 CRs (28%). After an average follow-up of 16.9 months, the response in 12 months was 74.6%, while the PFS at 12 months was 66.5% [71]. When Tislelizumab (anti-PD-1) was tested in the BGB-A317-203 study, in which 70 patients received 200 mg of tislelizumab intravenously every 3 weeks until disease progression or unacceptable toxicity, the ORR was 87.1% with a CR rate of 62.9%. After an average follow-up of 9.8 months, the estimated PFS in 9 months was 74.5% [72].

In a study that assessed the effect of pembrolizumab in six patients with metastatic EBV-positive GC, the ORR was reported to be 100% [73]. However, another study, which evaluated the effect of nivolumab, showed that only 25% of the advanced GC patients with positive EBV achieved an objective response [74].

HHV-8, the causative agent of Kaposi’s sarcoma, has a unique mechanism of driving tumor development. HHV-8 is not a classic oncogenic virus; the disease is an opportunistic tumor that responds to immune restoration, when possible [75]. Recently, some reports have suggested that ICIs may be effective in treating Kaposi’s sarcoma. For the first time, Delyon et al. (2018) reported partial response to nivolumab in two patients with severe endemic Kaposi’s sarcoma [76]. Saller et al. (2018) reported a case of advanced classic Kaposi’s sarcoma refractory to multiple lines of chemotherapy showing partial response to anti-PD-1 therapy with pembrolizumab. The investigation of immunotherapy in classic or endemic Kaposi’s sarcoma resulted in two phase II studies—one of nivolumab and ipilimumab (NCT03219671), and the other, a multicenter study of pembrolizumab (NCT03469804).

Merkel cell carcinoma (MCC), a polyomavirus-associated cancer, is an aggressive form of skin cancer. Probably, the most important oncogenic event in the development of MCC is the integration of the Merkel cell polyomavirus (MCPyV) into the host genome [77,78]. Investigations into the tumor microenvironment in MCC have highlighted the interaction between host-specific immune responses and the tumor’s escape mechanisms [79,80,81].

The efficacy and safety of avelumab (anti-PD-1) in MCC were assessed in a phase II clinical trial (JNCT02155647) including 88 immunocompetent patients with metastatic stage IV MCC, who were otherwise in a good general condition (Eastern Cooperative Oncology Group 0 or 1). The drug was used in a second-line environment, after failure of at least one conventional chemotherapy line. The primary results demonstrated the drug’s efficacy and safety, which were then confirmed using additional analyses during the two-year follow-up [82,83,84].

Although avelumab is the first approved drug in this scenario, other ICIs are also being evaluated and have shown promising results. A phase II study (CITN-09/KEYNOTE-017) evaluated pembrolizumab in 50 patients with metastatic MCC (43 with stage IV and 7 with stage III unresectable disease, who were systemic treatment virgins). The ORR was 56%, including 12 (24%) CRs and 16 (32%) PRs. The median PFS was 16.8 months [85]. Based on these results, the FDA granted accelerated approval of pembrolizumab for patients with locally advanced or recurrent metastatic MCC in December 2018. Case reports have suggested the effectiveness of nivolumab in treating metastatic MCC [86]. In addition, a phase 1/2 study (CheckMate 358) that investigated nivolumab in thirty-nine patients with cancers associated with advanced viruses, including MCPyV-positive and -negative MCC (who received two or fewer previous therapies) demonstrated that nivolumab induced radiographic tumor regressions in approximately half of the treated patients [87].

### 2.3. Antisense Oligonucleotides

Despite studies exploring the possibility for more than 20 years, it is only in the last 5 years that single-stranded oligonucleotides have become a modality of choice in the fields of precision medicine and targeted therapeutics. Single-stranded oligonucleotides are synthetic, short, modified RNA or DNA molecules that function in either a sequence-dependent (antisense oligonucleotides (ASOs) and immune stimulatory oligonucleotides) or tertiary structure-dependent (e.g., aptamers) manner. In 2020, Scharner and Aznarez published a review that highlighted the key applications of single-stranded oligonucleotides that function in a sequence-dependent manner when applied to modulate precursor (pre-)mRNA splicing, gene expression, and immune pathways. In virology, the main use of these single-stranded oligonucleotides is in the development of vaccines [88].

Epigenetic changes such as DNA methylation and modification of histones and noncoding RNAs (ncRNAs) influence gene expression without altering the DNA sequence. Although evidence shows that oncovirus infection is a precondition for cancer, it is known that viral infection alone is not enough to cause carcinogenesis, as carcinogenesis also depends on individual genetic variations and epigenetic changes. NcRNAs are abundant in the genome [89] and are divided into two subclasses: small ncRNAs (20–200 nucleotides) and long ncRNAs (lncRNAs; >200 nucleotides). LncRNAs are classified as exonic, intronic, overlapping, or intergenic, depending on their location in relation to the protein-coding genes [90]. LncRNAs regulate gene expression via cis and trans regulatory mechanisms. Recent advances in high-throughput sequencing have led to the identification of the biological functions of ncRNAs. These have been found to regulate several biological processes, including development, proliferation, repair, and differentiation of DNA damage [91,92]. Deregulation of lncRNAs has been linked to the development of cancers and other human diseases [93]. Therefore, lncRNAs have been studied as potential biomarkers and therapeutic targets for cancer caused by oncoviruses. He J et al. (2020) showed that in case of HPV, many lncRNAs are frequently deregulated and play crucial roles in cervical cancer. LncRNAs have also been found to influence cancer progression through wide regulation at the transcriptional, translational, and post-translational levels. Thus, detection of these specific roles may provide opportunities for early diagnosis of cervical cancer and identify new therapeutic opportunities in cervical cancer. A recent study conducted with HHV-8/KSHV demonstrated the feasibility of using carbon dots (Cdots) to deliver miRNA suppressors for targeting viral cancers and reported that using targeting miRNAs may provide a scientific basis for using antisense drugs against human cancers associated with oncogenic viruses [94]. For EBV, Murat et al. (2014) suggested alternative therapeutic strategies have focused on targeting the RNA structure within viral Open Reading Frame ORFs, by virus-encoded nuclear antigen 1 (EBNA1) mRNA; it was demonstrated that destabilization of G-quadruplexes using ASOs increases EBNA1 mRNA translation [95]. The explanatory mechanism for the pro-apoptotic activity of oncogenic latent membrane protein 1 (LMP1) of EBV being involved in the pathogenesis of NPC and lymphoma has also been suggested as a new therapeutic target to control tumor growth [96].

HCV treatment has proved to be successful and affordable with the use of these directly acting small-molecule drugs, which decrease the number of clear HCV infections, and consequently, the development of carcinoma. Conversely, since HBV is not susceptible to elimination using a similar approach, gene therapy for HBV infection remains a priority [97]. Billioud et al. (2016) demonstrated the efficacy of an antisense approach (alone or in combination with standard nucleoside therapy) against HBV in a preclinical setting, as assessed in terms of an efficient reduction in serum virus surface antigen (HBsAg), as well as viremia across different genotypes [98]. Antisense oligonucleotides have also been described in multiple preclinical studies as agents for chronic HBV infection treatment that can reduce the HBV viral load and the HBsAg and HBV replication serological marker (HBeAg) levels; however, challenges such as stability and intracellular delivery remain major hurdles for developing broadly applicable ASO-based HBV treatment [99].

Synthetic nucleic acid ligands (aptamers) have emerged as effective delivery vehicles for therapeutic oligonucleotides, including siRNAs (short interfering RNAs) [100].

Cancer-targeted therapy is an evolving treatment approach with great promise in increasing the effectiveness of cancer therapies. Delivery of therapeutic agents specifically to and in the desired tumor cells, using viral or nonviral targeting elements, is a promising therapeutic approach against cancer. Aptamers can be used as elements to target, replicate, and kill malignant cancer cells selectively, without affecting the surrounding healthy cells. Aptamers, on the other hand, are nonviral targeting elements that are single-stranded nucleic acids with high specificity, selectivity, and binding affinity in relation to their cognate targets. Aptamers have emerged as a new class of bioafinity-targeting elements that can be generated and molecularly manipulated to selectively bind to various targets, including proteins, cells, and tissues. Aptamer mediated to oncovirus therapies offers advantages such as target specificity, therapeutic payload, bioavailability of therapeutic agents at target sites, and minimization of systemic cytotoxicity. In a review article, Tan et al. (2016) emphasized the effective mechanisms for targeting to the location and issues of efficacy that affect the clinical applications of this technique [101].

### 2.4. RNA Interference

In 1998, the term RNA interference (RNAi) was first described by Fire et al. (1998), using cells from Caenorhabditis elegans [102]. The authors realized that the mechanism of this interference is not mediated by a simple antisense model, which was known until then as being an option to block endogenous gene expression. Instead, this interference is due to a unique phenomenon that displays a high degree of specificity, significant potency, and an unusual ability to cross cell boundaries. Later, it was described that RNAi may be present in a wide variety of organisms, such as fungi, plants, invertebrates, and vertebrates [103,104,105]. Activation of the RNAi response occurs in the presence of double-stranded RNA (dsRNA) in the cells. DsRNA, when inserted or synthesized within mammalian cells, is processed by an RNaseIII-like enzyme called Dicer. The resulting dsRNAs formed, which are about 21–23 nucleotides in length, are called siRNAs [106,107]. In addition, siRNA facilitates the construction of a multi-subunit RNA-induced silencing complex (RISC) that is roughly defined with cellular proteins containing nuclease and helicase activity, which together result in target recognition of intracellular mRNAs. Hydrolysis of RNA takes place internally in the homology region targeted by the siRNA, thus effectively leading to mRNA cleavage [108]. Over the years, this technique has been widely used in biology, virology, and cancer therapy for various disorders.

Hepatocarcinogenesis (HCC), as well as various types of cancer, is a multistage process that involves the accumulation of genetic or epigenetic changes deregulating oncogenes or tumor suppressors as well as cellular processes, such as cell proliferation, apoptosis, and angiogenesis, during initiation and progression of this disease. Taking into account the liver disease conditions and their genetic/epigenetic changes in human HCC, RNAi targets for the treatment of liver cancer have largely focused on HBV/HCV infection, the oncogenic and genes that regulate tumor growth and metastasis [109].

The first study that demonstrated that siRNA can trigger post-transcriptional repression of gene expression in vivo was performed in animals infected with the hepatitis B virus [110]. After this study, the first preclinical therapeutic application of siRNAs targeted against Fas in a mouse model of hepatitis, resulting in the protection of animals treated against liver damage [111]. Since then, several other studies have been published demonstrating the inhibitory effect of siRNA or shRNA on the tumor cell proliferation by suppressing the expression of oncogenes or other genes related to tumor growth [112,113,114]. For example, Vascular endothelial growth factor VEGF, a well-known target for cancer therapy, has been widely used as a therapeutic target for RNAi [115,116]. RNAi technology in cancer treatment has great advantages such as the use of small interfering RNA molecules that can be generated to target a large number of different genes related on a number of different signaling pathways. That is very important for a disease as complex as cancer. As RNAi can be easily delivered to the liver, HCC has been often investigated as a disease model for testing RNAi therapy.

In a study by Wang et al., RNAi using lentivirus was performed to inhibit the expression of 2A latent membrane (LMP2A) gene, in order to explore the effects of silencing LMP2A on the growth of a gastric carcinoma cell line associated with EBV (EBVaGC) in vitro. Lentivirus-mediated RNAi technology was used to silence the LMP2A gene in EBV-positive gastric carcinoma of the GT38 cell line [117].

Notably, for HPV, it has been reported that RNAi targeting of E7 or E6/E7 promotes the accumulation of TP53 and/or pRB, eventually leading to the induction of apoptosis and/or senescence in HPV16-positive cervical cancer cell lines (JIANG; MILNER, 2002 [118]; SIMA [119,120]), as well as, in HPV18-positive human cervical cancer cells [121,122]. A previous study showed that two siRNAs targeting the E6/E7 promoter and E7 transcripts produced E6 and E7 mRNA knockdown, increased TP53 protein levels, decreased CDKN2A (p16INK4A) protein levels, and inhibited studies on a new human cell line (SiHa )cell growth via apoptosis. siRNA was related to histone modification via histone methylation of H3–Lys9 [123,124].

Thus, combining siRNA with other treatment strategies may have great potential for the development of future therapies.

### 2.5. CRISPR-Based Gene Editing

Prokaryotes have evolved various defense mechanisms to withstand exposure to invading foreign nucleic acids [125]. CRISPR/Cas9 (clustered regularly interspaced short palindromic repeats/CRISPR-associated protein), an ancient antiviral system that has recently been discovered in bacteria, has shown tremendous potential as a precise, invariant genome-editing tool. CRISPR loci consist of a cluster of Cas genes, which encode Cas effector proteins and a “CRISPR matrix” [91]. The CRISPR matrix has repetition sequences, 21–48 bp in length, interspersed with variable sequences of similar length known as “spacers”, which are associated with foreign DNA sequences of previously found DNA elements (“protospaces”) [126]. New spacers are introduced in the CRISPR matrix by the action of some Cas effector proteins, after finding invasive DNA sequences. These spacers are like a “memory” to the organism against infectious agents found previously and are used to eliminate posterior foreign invasive genetic elements [127,128]. There are three stages in the silencing of viral aliens mediated by CRISPR/Cas: (i) acquisition and integration of spacer sequences in the CRISPR matrix; (ii) assembly of the antiviral endonuclease complex; (iii) silencing of foreign genetic elements via endonucleolytic cleavage using the CRISPR/Cas-based system, to activate host defenses or genetically modify viral genomes, thus providing novel, successful antiviral mechanisms and treatment modalities [129,130]. Although it was discovered as a genetic editing technique, CRISPR/Cas9 has been frequently used as an antiviral mechanism. Some studies have already used this technique to inhibit HBV, EBV, HPV, and even HSV-1 [131,132]. 

The ease with which the CRISPR/Cas9 system can be easily adapted to target any gene of interest, including pathology-causing genes, inspired several groups to develop the use of RNA-guided nucleases as a way of treating HBV infection (for review, see [133]). In 2014, Lin et al. were the first to assess the effectiveness of using CRISPR/Cas against HBV [134]. They evaluated eight HBV genotype-specific sgRNAs. Four of these combinations resulted in the inhibition of intracellular expression of HBcAg and HBsAg significantly, with suppression of up to 70% observed. Regarding HBV and cancer, Feng investigated the roles of lncRNA proliferating cell nuclear antigen pseudogene 1 (PCNAP1 )in the contribution of HBV replication through the modulation of miR-154/PCNA/HBV cccDNA signaling in hepatocarcinogenesis using CRISPR/Cas9, and as results the expression levels of PCNAP1 and PCNA were significantly elevated in the liver of HBV-infectious human liver-chimeric mice [135]. In addition to these, another study developed unique guide RNAs (sgRNAs) specific to the open reading frames, preS1/preS2/S, of the HBV genome and established HBsAg knockout HCC cell lines using the CRISPR/Cas9 system. They showed that knocking out HBsAg in HCC cell lines decreased HBsAg expression and significantly attenuated HCC proliferation in vitro, as well as tumorigenicity in vivo [136].

The DNA targeting capacity of the CRISPR/Cas9 system from Streptococcus pyogenes has been widely characterized for use against HCV. However, the potential therapeutic utility of CRISPR/Cas RNA targeting has not been investigated extensively. Sampson and colleagues identified a CRISPR/Cas9 system of the pathogenic bacterium, *Francisella novicide* (Fn), whose function involves endonucleolytic cleavage of endogenous bacterium transcription [137]. Interestingly, the Fn CRISPR/Cas9 system (FnCas9) is also able to direct DNA through the canonical pathway [138]. Upon targeting the HCV-negative RNA sequence, the FnCas9 system interferes with viral replication. While the questions on the exact mechanism by which FnCas9 manages to silence the viral genome still remain, this new RNA inactivation activity was added to the CRISPR/Cas9 functionality repertoire [139]. Studies that explain HCV infection and carcinogenesis, mainly using the CRISPR technique, are still few. Mitchell showed that liver-derived HVG2 cells, permissive to HCV, designed to constitutively express microRNA-122 (HepG2/miR-122 cells) have normal p53-mediated responses to DNA damage and that HCV replication in these cells potently suppresses p53 responses to etoposide, a DNA damage inducer, or nutlin-3, an inhibitor of p53 degradation pathways, thus demonstrating that HCV infection initiates p53 activation after DNA damage. This study suggests that persistent interruption of p53-mediated DNA damage responses may contribute to hepatocellular carcinogenesis in chronically infected individuals [140].

The Epstein–Barr virus (EBV) efficiently transforms primary human B cells into immortalized lymphoblast cell lines (LCLs). One study described the design and delivery of single-guide RNAs (sgRNAs) through lentiviral transduction of LCLs that stably express the Cas9 protein. The CRISPR/Cas9 edition performs the loss of LCL function, including knock-out of genes encoding proteins or exclusion of regulatory DNA elements, and can be adapted for large-scale screening approaches. In that study, they suggested that this editing technique could also be used on other B cell lines, including Burkitt’s lymphoma and large, diffuse lymphoma B cells, and is widely reproducible [141]. In another study, Wang and Quake investigated the application of CRISPR/Cas9 in the treatment of latent EBV virus infection in cells derived from Burkitt’s lymphoma patients with latent infection by the Epstein–Barr virus [142]. They designed several sgRNAs that targeted the location of the EBV genome responsible for viral structure, transformation, and latency. Cells derived from patients with latent EBV infection were treated with the constructed CRISPR/Cas9. After treatment with CRISPR/Cas9, the EBV genome was eliminated in a quarter of the cells, while half of the cells showed a decrease in viral load. This study demonstrated a 50% reduction in the number of copies of the viral genome [142].

In a study by Tso et al., a replication-compatible type 5 adenovirus was designed to generate a latency-associated nuclear antigen LANA-specific Cas9 system (Ad-CC9-LANA) in several KSHV latent target cells. The LANA gene, the latency-associated nuclear antigen, is absolutely required in the maintenance, replication, and segregation of KSHV episomes during mitosis, which makes this gene an ideal target for CRISPR/Cas9 editing. This study demonstrated that KSHV-infected epithelial and endothelial cells transduced with Ad-CC9-LANA underwent reductions in their KSHV episomal load, RNA LANA, and protein expression, though this effect was less profound in Boss Stem Cells BC3 cells, due to low infection efficiency of adenovirus type 5 in B cells. Thus, they concluded that the use of an adenovirus vector may confer potential in vivo applications of LANA-specific Cas9 against KSHV infection [143].

### 2.6. Other Gene Therapies and Treatments

Recently, various platforms have been developed for the genetic engineering of somatic and pluripotent stem cells. These include, in addition to CRISPR/Cas, zinc finger nuclease (ZFN) and TALENs. These platforms are being extensively explored, mainly in the field of biotechnology, and more recently, in first-phase clinical trials [144,145,146].

#### 2.6.1. Zinc Finger Nuclease

ZFN is a chimeric protein designed to recognize and cleave specific sequences of nucleotides in genomic DNA, thus enabling site-directed mutagenesis. ZFNs are constructed as modules of two or more zinc finger domains linked to the cleavage domain of a nuclease, usually the restriction enzyme FokI. ZFN displays catalytic activity only when it is dimerized, so its use for genomic engineering imposes the need to design pairs of ZFNs that are capable of binding inverted sequences that are in opposite strands of DNA [142]; [69]. In addition, the carboxyterminal ends of the ZFNs need to be approximately 4–7 base pairs apart from each other, to allow dimerization of the FokI cleavage domains. The main objective of therapy using Zinc Finger Nuclease ZNF is to develop personalized protein designs that are engineered to recognize and cleave double-stranded DNA at specific locations in the genome [147].

The therapeutic application of ZFN technology requires highly specific engineering. Recently, new applications for ZFN technology have been developed through the creation of new specific cleavage domains for FokI. Genome editing using ZFN has great potential to assist in the discovery of new medicinal drugs or in direct application as nucleases for therapeutic purposes [148,149]. Sequence-specific nucleases may also be used to disable genes that encode pathology-causing proteins [150].

ZFNs are promising tools for the treatment of some diseases. For treatment of HPV infection, studies have focused on editing of the viral oncoproteins E6 and E7. ZFNs targeting HPV16 and 18 induced in vitro E7 interruption and inhibition of type-specific and efficient growth and apoptosis of HPV-positive cells. These targeted ZFNs may be possible new therapeutic agents for the treatment of HPV-related cervical cancer [151,152,153].

The potential advantage of a ZFN approach is fully penetrating and hereditary genetic silencing, which persists throughout the life of the cell [154,155]. The disadvantage of this technique is its complexity, in addition to toxicity and DNA breaks at unwanted places [156].

#### 2.6.2. Transcription Activator-Like Effector Nucleases

The TALEN technique emerged after the development of the ZFN technology, as an alternative for editing the human genome. Like ZFNs, TALENs also make use of cellular repair mechanisms in a certain region of the genome or decrease the expression of a gene. Although TALENs also make use of the restriction endonuclease domain of the Fok I enzyme, similar to ZFNs, their TALEN binding domain is composed of repetitions, similar to the transcription activator-like promoters (TALE) protein class, which were discovered in a group of bacteria [156,157,158]. Each repetition has two adjacent amino acids that are called variable di-residues, RVDs, which confer specificity to one of the four base pairs of DNA. The specificity of a TALE’s DNA binding is determined by its number of repetitions and the sequence of the RVD, in which the number of repetitions determines the length of the target sequence.

Currently, TALENs have proven to be an effective tool for antiviral therapy because of their low toxicity and the possibility of being designed to adapt to an antiviral strategy [159]. Therefore, TALENs are likely to become part of a new approach for the treatment of various infections.

Studies have tested the effectiveness of in vitro use of TALEN in blocking lytic replication of HCMV in infected mouse cells. Administration of TALEN plasmids resulted in a significant reduction in the number of gene copies, which is key to viral latency. The results suggested that TALENs are an effective strategy for cleaning up latent HCMV in animals [160]

Other studies have demonstrated the application of TALENs in HPV. Shankar S et al. indicated that TALENs selected in this study were effective in eliminating HPV, as the TALEN-mediated version of E7 led to cell death [161]. Other studies conducted with an intention to edit E7 using TALEN have also obtained similar results [152,162,163,164].

The combination of nanoparticle-based delivery systems and genome editing tools is a powerful and promising strategy for cervical cancer therapy [153].

## 3. The Future of Gene Therapy for Oncogenic Viruses

Despite rapid advances, gene editing is still a challenge, mainly due to the difficulty of some variables such as gene delivery, specificity, and effectiveness. The strategies applied in preclinical in vivo trials can be conducted in a controlled manner, are capable of efficient engineering using various methods, and can be a more direct way to bring precision gene-editing medicine to the clinic [165]. However, in vivo, the strategies also face the challenge of uncontrolled off-target events, due to the disadvantages of each technique (Table 1). Thus, it is important to exercise caution and evaluate the benefits of any technique against its possible risks. In order to fully understand the long-term effects of potential new treatments involving ideal and accurate gene therapy, it is necessary that the preclinical studies are carried out in a thorough and systematic manner.

## Figures and Tables

**Table 1 viruses-13-00822-t001:** Comparison of the advantages and disadvantages of clustered regularly interspaced short palindromic repeats (CRISPR)/Cas-9, ZNF, and transcription activator-like effector nucleases (TALEN) gene therapies.

Application	Advantages	Disadvantages
CRISPR/Cas9	-Simple and staggered system; -Low cost; -Double-stranded DNA breaks can be made either using homology-directed repair HDR or using the nonhomologous terminal bond;	-Induced mutations; -Use of viral vectors; -Unexpected genetic rearrangements (deletions and insertions) of DNA;
ZNF	-Double-stranded DNA breaks can be made either using HDR or using nonhomologous terminal bond; -It can be used in pluripotent or somatic stem cells;	-High cost; -Laborious; -It needs specific regions and rich in guanine; -Induced mutations;
TALEN	-Double-stranded DNA breaks can be done either using either HDR or the nonhomologous terminal bond -Adaptable	-Difficult plasmidial construction, due to the size; -Fok I domain used may not be specific; -Induced mutations

## Data Availability

Not applicable.

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
