# Peer review of "Immunotherapy and Gene Therapy for Oncoviruses Infections: A Review"

_viruses, 2021, doi:10.3390/v13050822_

Round 1

Reviewer 1 Report

The authors reviewed the treatment of oncovirus infections with oncolytic immunotherapy, immune checkpoint inhibitors, antisense oligonucleotides, RNA interference, CRISPR-based gene editing, zinc finger nuclease, and transcription activator-like effector nucleases.

This review is characterized by its focus on immunotherapy and gene therapy for malignancies caused by oncoviruses and can be valuable. However, there are some points to reconsider, as described below.

Title: “Immunotherapy and gene therapy for oncoviruses: A review” can be “Immunotherapy and gene therapy for oncovirus infections: A review”.

Introduction: The authors stated that there are currently eight well-accepted oncoviruses, including EBV, HTLV-1, HBV, HCV, HPV, HHV-8, MCPyV, and HCMV. What are the malignant tumors caused by HCMV? The authors should present papers demonstrating the oncogenicity of HCMV. In sections 2.4, 2.5, 2.6.1, and 2.6.2, the authors described anti-HIV-1 siRNAs (page 8, lines 366-377), elimination of internal integrated HIV-1 genes in infected host cells with CRISPR/Cas9 (page 10, lines 450-461), and genetic alteration of CCR5 using ZFN (page 12, lines 526-534) and TALEN (page 12, liens 565-570). If HIV-1 itself is not an oncovirus, these descriptions should be reconsidered. To retain these descriptions, the authors should change the title and purpose of this paper.

Page 2, line 81: Theses references [14-17] do not include EBV-related lymphoma. The authors need to add other relevant references. As an example, Taylor-GS et al. reported a recombinant modified vaccinia Ankara vaccine encoding EBV target antigens in a phase I trial in UK patients with EBV-positive nasopharyngeal carcinoma (Clin Cancer Res 2014, 20:5009).

Page 3, liens 93: Reference 24 used NDV but not VSV as the oncolytic virus. This needs to be rectified.

Page 5, lines 217-233: The authors described the effects of ICIs on classical Hodgkin lymphoma. It has not been confirmed whether all classical Hodgkin lymphomas are associated with EBV. Therefore, it is necessary to state whether the patients examined in the cited references have EBV-related Hodgkin lymphoma.

It is better to cite a paper reviewing CRISPR/cas9 technology for combating human viral infections (de Buhr et al. Curr Opin Immunol 2018, 54:123).

References should be changed, especially the author’ name, to suit the style of the journal.

Author Response

The authors reviewed the treatment of oncovirus infections with oncolytic immunotherapy, immune checkpoint inhibitors, antisense oligonucleotides, RNA interference, CRISPR-based gene editing, zinc finger nuclease, and transcription activator-like effector nucleases.

This review is characterized by its focus on immunotherapy and gene therapy for malignancies caused by oncoviruses and can be valuable. However, there are some points to reconsider, as described below.

Title: “Immunotherapy and gene therapy for oncoviruses: A review” can be “Immunotherapy and gene therapy for oncovirus infections: A review”.

The suggested was accepted.

Introduction: The authors stated that there are currently eight well-accepted oncoviruses, including EBV, HTLV-1, HBV, HCV, HPV, HHV-8, MCPyV, and HCMV. What are the malignant tumors caused by HCMV? The authors should present papers demonstrating the oncogenicity of HCMV.

The manuscripts below were added to demonstrate the oncogenicity of HCMV.

Herbein G. The Human Cytomegalovirus, from Oncomodulation to Oncogenesis. Viruses. 2018 Aug 3;10(8):408. doi: 10.3390/v10080408. PMID: 30081496; PMCID: PMC6115842.

Cobbs C. Cytomegalovirus is a tumor-associated virus: armed and dangerous. Curr Opin Virol. 2019 Dec;39:49-59. doi: 10.1016/j.coviro.2019.08.003. Epub 2019 Sep 14. PMID: 31525538.

In sections 2.4, 2.5, 2.6.1, and 2.6.2, the authors described anti-HIV-1 siRNAs (page 8, lines 366-377), elimination of internal integrated HIV-1 genes in infected host cells with CRISPR/Cas9 (page 10, lines 450-461), and genetic alteration of CCR5 using ZFN (page 12, lines 526-534) and TALEN (page 12, liens 565-570). If HIV-1 itself is not an oncovirus, these descriptions should be reconsidered. To retain these descriptions, the authors should change the title and purpose of this paper.

We agree with the reviewer that the focus is oncogenic viruses, which is why HIV reviews have been removed from the text.

Page 2, line 81: Theses references [14-17] do not include EBV-related lymphoma. The authors need to add other relevant references. As an example, Taylor-GS et al. reported a recombinant modified vaccinia Ankara vaccine encoding EBV target antigens in a phase I trial in UK patients with EBV-positive nasopharyngeal carcinoma (Clin Cancer Res 2014, 20:5009).

The reference from Taylor-GS et al. reported a recombinant modified vaccinia Ankara vaccine encoding EBV target antigens in a phase I trial in UK patients with EBV-positive nasopharyngeal carcinoma (Clin Cancer Res 2014, 20:5009) and Taylor GS and Long HM, 2015 were added.

Page 3, liens 93: Reference 24 used NDV but not VSV as the oncolytic virus. This needs to be rectified.

The error was rectified.

Page 5, lines 217-233: The authors described the effects of ICIs on classical Hodgkin lymphoma. It has not been confirmed whether all classical Hodgkin lymphomas are associated with EBV. Therefore, it is necessary to state whether the patients examined in the cited references have EBV-related Hodgkin lymphoma.

As suggested more information was added about EBV status of the patients. Page 5.

It is better to cite a paper reviewing CRISPR/cas9 technology for combating human viral infections (de Buhr et al. Curr Opin Immunol 2018, 54:123).

As suggested the reference from de Buhr et al. Curr Opin Immunol 2018, 54:123 was added.

References should be changed, especially the author’ name, to suit the style of the journal.

The references were configured using the endnote program as requested by the journal. Below are images that prove that the correct model (MDPI Chicago) was followed correctly.

Authors review cancers caused by viruses (oncoviruses) and different therapies against them.

Line 53: The sentence “Currently, gene therapy is used for oncolytic viruses” does not seem to make any sense. How is gene therapy used for oncolytic viruses?  Please write the concept in other words.  Maybe it refers to oncoviruses, which is different from oncolytic.  If it refers to oncolytic, the oncolytic virus concept needs to be introduced before this sentence.

We agreed had a misunderstood between oncolytic viruses and oncoviruses. This paragraph refers to oncovirus. The word was changed. Page 2

Line 113 to 137:  This section is very misleading. Authors talk about treating AIDS patients  (HIV-infected) with ICIs. Although AIDS is caused by a virus, it has no relation to a cancer induced by a virus. It is quite misleading to mix different concepts. ICIs work different in HIV-infected patients because AIDS creates immune suppression. This immune suppression also contributes to a greater cancer incidence. However, the paper should focus on how to treat virus-induced tumors. Is the tumor induced by HIV?  Authors mention that “A systematic review conducted by Cook et al. (2019) to summarize the effectiveness of ICI therapy in HIV-positive cancers”  . Are these really HIV-positive cancers?? Or cancers in HIV-positive patients, which is completely different.

We agree and appreciate the suggestion, the sentence has been rewritten and was added more detail.  Page 5 and 6.

Section 2.2: Besides enumerating treatments for tumors caused by viruses, authors should provide more insight in the mechanisms of action of a drug and the mechanism of oncogenesis by the virus. Why is appropriate to use ICIs for a virus-caused cancer? How viruses modulate checkpoint inhibitors in lymphocytes and their ligands in tumors? Is there a cancer-causing virus that evades the immune system using ICIs more than others? Is there any clinical evidence that viral antigens also tumor antigens in tumors caused by viruses?

As suggested, A brief discussion was added in the page 3.

Paragraph starting in line 333: why the concept of oncolytic viruses is mixed with aptamers here?

The sentence was wrong and was rewritten.  Page 2 (Line 351 to 362)

Paragraphs starting in lines 366 , 379, and 404: Here RNAi is used as antiviral therapy. The paper should mainly focus on antitumor therapy, otherwise it jumps from one concept to another one, and it becomes confusing.  Is this paper about antivirals? Even these viruses cause cancer, antiviral and antirumoral are two very different areas. Where is described the antirumoral potential of these antiviral siRNAs?

We agree and change as suggested. A paragraph was added describing the applications of RNAi in cancer related to HCV and HBV, since for EBV and HHV-8 we have already described it. Page 9

The same can be said for CRISPR (section 2.5) when used only as antivirals. Comments should mainly focus on antitumoral activity secondary to its antiviral activity, which should be the common theme of the review. Of course, preventing viral infection with these viruses prevents futures possible tumors, but dealing with prevention and treatment in the same paper is again confusing. Otherwise, I would suggest separating the text in two sections: antivirals to prevent cancer, and cancer treatment.

As in the previous topic, the information has been modified. Studies related to cancer and the CRISPR technique have been added to the paragraphs on HCV, HBV and EBV. For HHV-8 it had already been described, considering that LANA is considered one of the most important studied in cancer. Page 11

Reviewer 2 Report

Authors review cancers caused by viruses (oncoviruses) and different therapies against them.

Line 53: The sentence “Currently, gene therapy is used for oncolytic viruses” does not seem to make any sense. How is gene therapy used for oncolytic viruses?  Please write the concept in other words.  May be it refers to oncoviruses, which is different from oncolytic.  If it refers to oncolytic, the oncolytic virus concept needs to be introduced before this sentence.

Line 113 to 137:  This section is very misleading. Authors talk about treating AIDS patients  (HIV-infected) with ICIs. Although AIDS is caused by a virus, it has no relation to a cancer induced by a virus. It is quite misleading to mix different concepts. ICIs work different in HIV-infected patients because AIDS creates immune suppression. This immune suppression also contributes to a greater cancer incidence. However, the paper should focus on how to treat virus-induced tumors. Is the tumor induced by HIV?  Authors mention that “A systematic review conducted by Cook et al. (2019) to summarize the effectiveness of ICI therapy in HIV-positive cancers”  . Are these really HIV-positive cancers?? Or cancers in HIV-positive patients, which is completely different.

Section 2.2: Besides enumerating treatments for tumors caused by viruses, authors should provide more insight in the mechanisms of action of a drug and the mechanism of oncogenesis by the virus. Why is appropriate to use ICIs for a virus-caused cancer? How viruses modulate checkpoint inhibitors in lymphocytes and their ligands in tumors? Is there a cancer-causing virus that evades the immune system using ICIs more than others? Is there any clinical evidence that viral antigens also tumor antigens in tumors caused by viruses?

Paragraph starting in line 333: why the concept of oncolytic viruses is mixed with aptamers here?

Paragraphs starting in lines 366 , 379, and 404: Here RNAi is used as antiviral therapy. The paper should mainly focus on antitumor therapy, otherwise it jumps from one concept to another one, and it becomes confusing.  Is this paper about antivirals? Even these viruses cause cancer, antiviral and antirumoral are two very different areas. Where is described the antirumoral potential of these antiviral siRNAs?

The same can be said for CRISPR (section 2.5) when used only as antivirals. Comments should mainly focus on antitumoral activity secondary to its antiviral activity, which should be the common theme of the review. Of course, preventing viral infection with these viruses prevents futures possible tumors, but dealing with prevention and treatment in the same paper is again confusing. Otherwise, I would suggest separating the text in two sections: antivirals to prevent cancer, and cancer treatment.

Author Response

Line 53: The sentence “Currently, gene therapy is used for oncolytic viruses” does not seem to make any sense. How is gene therapy used for oncolytic viruses?  Please write the concept in other words.  Maybe it refers to oncoviruses, which is different from oncolytic.  If it refers to oncolytic, the oncolytic virus concept needs to be introduced before this sentence.

We agreed had a misunderstood between oncolytic viruses and oncoviruses. This paragraph refers to oncovirus. The word was changed. Page 2

Line 113 to 137:  This section is very misleading. Authors talk about treating AIDS patients  (HIV-infected) with ICIs. Although AIDS is caused by a virus, it has no relation to a cancer induced by a virus. It is quite misleading to mix different concepts. ICIs work different in HIV-infected patients because AIDS creates immune suppression. This immune suppression also contributes to a greater cancer incidence. However, the paper should focus on how to treat virus-induced tumors. Is the tumor induced by HIV?  Authors mention that “A systematic review conducted by Cook et al. (2019) to summarize the effectiveness of ICI therapy in HIV-positive cancers”  . Are these really HIV-positive cancers?? Or cancers in HIV-positive patients, which is completely different.

We agree and appreciate the suggestion, the sentence has been rewritten and was added more detail.  Page 5 and 6.

Section 2.2: Besides enumerating treatments for tumors caused by viruses, authors should provide more insight in the mechanisms of action of a drug and the mechanism of oncogenesis by the virus. Why is appropriate to use ICIs for a virus-caused cancer? How viruses modulate checkpoint inhibitors in lymphocytes and their ligands in tumors? Is there a cancer-causing virus that evades the immune system using ICIs more than others? Is there any clinical evidence that viral antigens also tumor antigens in tumors caused by viruses?

As suggested, A brief discussion was added in the page 3.

Paragraph starting in line 333: why the concept of oncolytic viruses is mixed with aptamers here?

The sentence was wrong and was rewritten.  Page 2 (Line 351 to 362)

Paragraphs starting in lines 366 , 379, and 404: Here RNAi is used as antiviral therapy. The paper should mainly focus on antitumor therapy, otherwise it jumps from one concept to another one, and it becomes confusing.  Is this paper about antivirals? Even these viruses cause cancer, antiviral and antirumoral are two very different areas. Where is described the antirumoral potential of these antiviral siRNAs?

We agree and change as suggested. A paragraph was added describing the applications of RNAi in cancer related to HCV and HBV, since for EBV and HHV-8 we have already described it. Page 9

The same can be said for CRISPR (section 2.5) when used only as antivirals. Comments should mainly focus on antitumoral activity secondary to its antiviral activity, which should be the common theme of the review. Of course, preventing viral infection with these viruses prevents futures possible tumors, but dealing with prevention and treatment in the same paper is again confusing. Otherwise, I would suggest separating the text in two sections: antivirals to prevent cancer, and cancer treatment.

As in the previous topic, the information has been modified. Studies related to cancer and the CRISPR technique have been added to the paragraphs on HCV, HBV and EBV. For HHV-8 it had already been described, considering that LANA is considered one of the most important studied in cancer. Page 11

Round 2

Reviewer 1 Report

The authors responded appropriately to the points raised by the reviewer.

Reviewer 2 Report

None